# Evaluation of Mechanical and Wear Properties of Al 5059/B$_4$C/Al$_2$O$_3$ Hybrid Metal Matrix Composites

Uppu Pranavi [1], Pathapalli Venkateshwar Reddy [1,*], Sarila Venukumar [1] and Muralimohan Cheepu [2,*]

1   Department of Mechanical Engineering, Vardhaman College of Engineering, Hyderabad 501218, India; upranavi908@gmail.com (U.P.); venukumar24@gmail.com (S.V.)
2   Department of Materials System Engineering, Pukyong National University, Busan 48513, Korea
*   Correspondence: mr.pvreddy@gmail.com (P.V.R.); muralicheepu@gmail.com (M.C.)

**Abstract:** There is a developing interest in efficient materials in automobile and aerospace fields that involves the improvement of metal matrix composites (MMCs) with great properties which incorporate higher strength, hardness and stiffness, better wear and destructive resistance along with better thermal properties. This work deals with the evaluation of the mechanical and wear properties of the newly developed hybrid MMC of Al 5059/B$_4$C/Al$_2$O$_3$ produced by stir casting method. The main aim of the work was to evaluate the mechanical properties of various MMCs fabricated with various weight proportions of ceramic particles (B$_4$C and Al$_2$O$_3$). An increase in the tensile strength and the surface hardness was observed with the increase in the ceramic particles but there was a decrease in the percentage of elongation of the specimen. An increase in the ceramic content in the composite samples made the composite sample brittle (composite) from ductile (base metal).

**Keywords:** Al-5059; ceramic materials; HMMC; hardness; tensile strength; wear; SEM

## 1. Introduction

The increasing demand for fuel-efficient vehicles in the automotive industry along with the reduction in energy consumption and air pollution took a step forward with the development of MMCs. MMCs are the engineered combination of metal matrix material and reinforcement particles, and have upgraded features like as high strength, very light weight, stiffness, excellent wear resistance and oxidation, and exhibit better corrosion resistance. All these tailored properties increase the use of MMCs in marine, aviation, electronics, nuclear and automobile industries. Hybrid metal matrix composites (HMMCs) are also a wide area of research at present [1–3]. HMMCs exhibit fracture toughness and the ability to withstand corrosion conditions at high temperatures. All these properties are easily achievable with MMCs/HMMCs of aluminum rather than other light-weight magnesium and monolithic titanium alloys. Additionally, the low-cost fabrication, high strength and corrosion resistance of aluminum make wide usage of the metal in many engineering applications practically. The composites made of Al and Al$_2$O$_3$ have efficient mechanical as well as tribological properties and as such they can be widely used in motor blocks and crank bearings, which improves the wear resistance [4–7]. Fatigue resistance, which is the most important and essential property for automotive applications, is obtained by using Al-MMCs. An increase in the mechanical strength and wear resistance of aluminum can be obtained when ceramic particles, like B$_4$C, ZrSiO$_4$, etc., are used for reinforcement. The composites made from particulate metal matrix comprise isotropic properties. The mechanical properties of the metal matrix composite depend on some important factors that comprise the size and weight fraction of the particulates and the method of developing the composite [8–11].

The fabrication technique adopted in the present work is considered on the basis of the work done by Bansal and Saini [12]. In their work, the stir casting process is considered simple, flexible and economical. Fabricated specimens of Al359 alloy with

variable wt% of SiC and SiC/graphite were investigated for tensile strength, wear resistance and surface hardness. The specimen with higher wt% of SiC/Gr exhibited improved wear and hardness properties, superior tensile strength and a continuous decrease in elongation and ductility. Reddy et al. [13] studied the influence of the substitution of carbides of silicon and titanium to the metal matrix of Al5052 and observed the enhancement of the mechanical properties of the aluminum alloy. The increase in the wt% of the TiC has gradually reduced the wear of the composite. A steady dissemination of carbide particles and a reduction in voids and grooves are observed from the SEM images. Reddy et al. [14] correlated the experimental work done by using BBD (Box–Behnken design) of RSM with the predicted results, verified with ANOVA, to obtain the optimum specific wear rate of Al6063/TiC MMC. Daniel et al. [15–17] explained clearly the experimental work and the procedure for identifying the optimal conditions for the improved behavior of composites like Al5059/SiC/MoS$_2$. Several of the suggested control factors such as the weight percentage (wt%), sliding speed and load are adopted in the present work. The load, wt% of SiC and sliding velocity influence the wear resistance of the composites as revealed from the research work. The Taguchi technique was applied and 27 tests conducted on Al5059/SiC MMC, and the optimum conditions were analyzed using ANOVA. The study included the stir casting process for the fabrication of the composites and the use of SEM images to observe the worn-out surface of three different fabricated specimens. The results depicted that out of samples of 5, 10 and 15 wt% of SiC in aluminum alloy 5059, the specimen with highest wt% of SiC showed dominant tribological behavior and better wear resistance at a particle size of 10 μm. The maximum material removal rate and minimum surface roughness was also obtained by the best combination of various parameters.

Joseph et al. [18] reviewed various processes and suggested stir casting as the best and most economical method to fabricate the composites. The study included obtaining the optimized process parameters to improve the behavior of the composite formed using Al7075 as base metal and various wt% of SiC and TiB$_2$ as reinforcement particles. Lawate et al. [19] studied the process parameters influencing the stir casting process to observe the behavior of alloys of aluminum when reinforced with Al$_2$O$_3$. Mohan et al.'s [20] research work included the analysis of several mechanical properties of Al6061/B$_4$C/Gr composite in which the constituents are added based on constant volume ratio. With the description of the characteristics of the aluminum alloy, carbide and graphite used, the study explained the improvement in hardness and toughness with the inclusion of B$_4$C and the increase in tensile strength with the proportionate increase in the graphite wt%. James et al. [21] studied the hardness, tensile strength and tribological properties of an Al6061/ZrO$_2$/Al$_2$O$_3$ composite. The specimens were fabricated using the stir casting method. The study reported an increase in the tensile strength, hardness and wear resistance along with the dissimilate distribution of the particulate constituents. Ajith et al. [22] observed that an artificial neural network model gives most convincing results to obtain the required properties of the composites by forecasting the optimal input parameters. The work was carried out on machined composite specimens of Al5052/SiC with a constant 2% of MoS$_2$. Suraya et al. [23] explained the effect on composite materials due to the variation of the wt% of reinforcement particles in the aluminum alloy. It was observed that there was gradual enhancement in the properties of the aluminum alloy composite if the wt% of TiC was between 0 and 10. Further increases of wt% of TiC resulted in a diminution of various significant properties of the alloy. The research work on MMCs exhibited various fabrication processes to develop the composite specimens and various analysis techniques to obtain optimized parameters that enhance several properties of the base metal for their application in numerous engineering fields [24]. Several researchers worked on titanium-based composites as well [25–28]. Zherebtsov et al. [25] explained the effect of hot rolling on the mechanical properties of the prepared titanium-based MMCs. Their work concluded that there is a substantial improvement in ductility at elevated temperature of up to ~12% elongation. The increase in ductility of the prepared MMCs is due to the presence of a more stable α phase in the hot rolled condition.



Koo et al. [26] worked on TiB-reinforced titanium alloy composites fabricated using a spark plasma sintering process. The effect of TiB's whiskers aspect ratio on the strengthening efficiency was studied in their work. Finally, the work concluded that the 58 aspect ratio whiskers obtained better strengthening efficiency when compared to the other aspect ratio whiskers. In another work of Chen et al. [27], the authors worked on the aspect ratio effect with regard to the strengthening of an aluminum-based metal matrix composite reinforced with carbon nano tubes. The work concluded that the prepared composites obtained better strength irrespective of the CNT aspect ratio.

The improvement in the physical, mechanical and tribological properties of different aluminum alloys depends on the percentage of reinforcement particles in the fabricated composite. Based on the above literature, the present work deals with the evaluation of the tensile strength, hardness and wear resistance of fabricated hybrid MMCs. Aluminum oxide improves physical and mechanical properties like tensile strength and hardness, whereas boron carbide improves the erosion resistance of the composite sample. The main aim of the present work is to fabricate hybrid MMCs to improve the dry sliding wear resistance and mechanical properties. The present study includes the evaluation of the effect of wear parameters like sliding speed, load and reinforcement content ($B_4C$ and $Al_2O_3$) on wear properties. Bonding among the particles and matrix material was studied by scanning electron microscopy of the fractured and wear-tested MMC. Based on the results obtained, that the developed composites can be used as an innovative standpoint in enhancing the properties of composites for aerospace and automotive applications.

## 2. Materials and Methods

### 2.1. Fabrication of Composite

The metal matrix composite (MMC) was fabricated by using an ultrasonic-probe-(USP)-assisted stir casting technique. The fabrication of the composite specimen was carried out using Al-5059 as the metal matrix material and the ceramic materials $Al_2O_3$ and $B_4C$ as reinforcement particles. The chemical composition of Al-5059 is shown in Table 1. From previous research, it was understood that there is an improvement in the physical and mechanical properties of composite materials when the percentage of reinforcement particles in a composite material is below 20% [24]. However, this results in a decrease in some properties, like ductility, making the composite unsuitable for some manufacturing processes, like forming. Thus, to keep the composites able to be used in any manufacturing methods, the maximum weight percentage (wt%) of the reinforcement material is 15% in the present research work. In this work, the wt% of reinforcement particles in each of the three fabricated composites was 5%, 10% and 15%. The fabrication procedure for HMMCs has been explained clearly in the literature [24,29,30]. The selected reinforcements were preheated to 500 °C initially. The matrix material Al-5052 was stir cast to 800 °C; once the matrix material is completely converted to liquid form, the required amount of reinforcement quantity (pre-heated) was added into the matrix by continuously stirring and sonicating from the bottom. The three specimens fabricated by varying the wt% of ceramic particles and the metal matrix are tabulated and shown in Table 2. The fabricated composite specimen samples are shown in Figure 1.

**Table 1.** Chemical composition of Al5059.

| Element | Si | Fe | Cu | Mg | Ti | Mn | Cr | Al |
|---|---|---|---|---|---|---|---|---|
| Weight Percentage | ≤0.45 | ≤0.50 | ≤0.25 | 5–6 | ≤0.20 | 0.60–1.2 | ≤0.25 | 89.8–94 |

**Table 2.** wt% of the composite materials in each specimen.

| Specimen Designation | Al5059 (wt%) | $B_4C$ (wt%) | $Al_2O_3$ (wt%) |
| --- | --- | --- | --- |
| SP 1 | 95 | 2.5 | 2.5 |
| SP 2 | 90 | 5 | 5 |
| SP 3 | 85 | 7.5 | 7.5 |

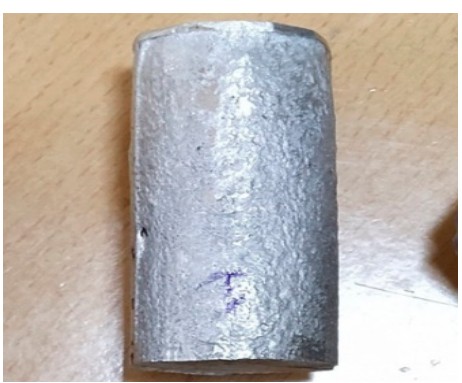

**Figure 1.** Fabricated hybrid composite sample using stir casting process.

### 2.2. Investigation on Mechanical Properties

The hardness test and the tensile test for determining the mechanical properties were conducted on three fabricated HMMC specimens. The procedures adopted to conduct these mechanical tests are clearly explained below.

### 2.3. Hardness Test

The surface of the composite specimens was polished with emery paper. The specimens were then used for obtaining the Brinell's hardness number using a Brinell hardness tester. The three composite samples were tested based on ASTM E384 [31] standards.

### 2.4. Tensile Test

The tensile strength of the HMMC specimens was tested using a computer-operated universal tensile machine, (UTM) INSTRON-3369, with a capacity of 50 kN. A crosshead speed of 0.5 mm/min at room temperature was applied to the composite specimens. The tensile test on the composite specimens was conducted as per ASTM E8 [32] standards.

### 2.5. Wear Test

The procedure for the wear test has been explained in detail in the literature [24]. The wear test was performed with a DuCom-made Pin-on-Disc tribometer. The ASTM G99 standards [33] were used as the basis for the wear test in the present work. A disk of track diameter 60 mm made with EN-32 steel was used as a counter-face for testing the specimen. Three specimens were tested by varying parameters like load and sliding speed. Load was varied from 10 N to 30 N with an increment of 10 N, whereas speed was varied from 500 rpm to 700 rpm with an increment of 100 rpm. The specific wear rate (SWR) was calculated by using Equation (1).

$$\text{SWR} = \frac{\Delta W}{\rho \times F_n \times S_d} \tag{1}$$

where $\Delta W$ is the weight loss, $\rho$ is the density of the specimen, $F_n$ is the normal load and $S_d$ is the sliding distance.

## 3. Results and Discussion

The tensile test, hardness test and the wear test results are precisely explained below. The impact of wear test parameters like applied load, sliding speed and reinforcement content on the friction behavior of HMMC was studied. Meanwhile, as the study is related to tribology, the specific wear rate and coefficient of friction were taken as structure responses. Accordingly, the effect of the tribo-testing conditions on the friction behavior of Al 5059/$B_4C$/$Al_2O_3$ is studied.

### 3.1. Brinell Hardness Test Results

Brinell's hardness test was performed on three HMMC specimens and the tested samples are shown in Figure 2. The mean of the hardness value of the tested specimens is shown in Figure 3. From the results shown in the Figure 3, the hardness value was greater for the sample SP3, which had a higher content of $B_4C$ and $Al_2O_3$. The tough interfacial bond between the metal matrix material (Al-5059) and the ceramic particles ($B_4C$ and $Al_2O_3$) was the cause for the high hardness value of the specimen, SP3. Additionally, the ceramic particle, $Al_2O_3$, was the hardest material. This property of $Al_2O_3$ could be one of the main reasons for the increase in the hardness of the composite sample SP3.

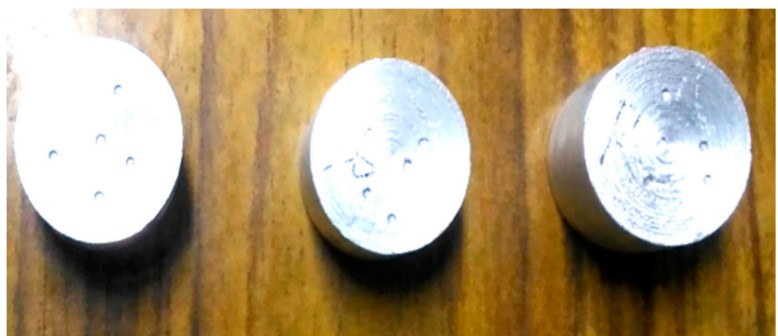

**Figure 2.** Hybrid composite specimens after hardness test.

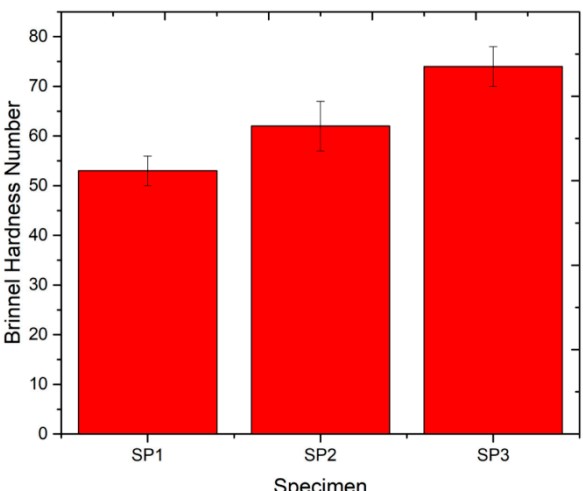

**Figure 3.** Brinell hardness number (BHN) for tested composite specimens.

From the literature, it was clearly observed that the increase in the ceramic content in the composite specimen increased the hardness of the composite specimen [13,14,24]. Bonding between the ceramic and matrix improved the mechanical properties. Similar results have been observed in the literature and the inference drawn from SEM images (see Section 4.1) is also in line with the hardness results. The inclusion of ceramic particles into the matrix material improved the strength; likewise, the ductility of the composite material

decreased and the composite became brittle. From the physics point of view, the hardness of the brittle material was higher than the ductile material, since the deformation of the material is less in the brittle material. The presence of $B_4C$ and $Al_2O_3$ in the interaction makeup up of particles and decreased vacancy content are the reasons for the increased hardness [13]. When the load is applied, reinforcement stops the movement of dislocations, resulting in an increased hardness. Furthermore, when the wt% of $B_4C$ and $Al_2O_3$ reaches its maximum, there is a risk of $B_4C$ and $Al_2O_3$ particle agglomeration, which lowers the interfacial adhesion across reinforcement and metal matrix, lowering the composites' surface hardness.

### 3.2. Tensile Test Results

The tensile test was carried out on the three fabricated specimens. The results of the tensile test are shown in Figure 4, from which it is evident that the tensile strength was higher in SP3, which had a higher percentage of ceramic particles. The wt.% of ceramic particles in this specimen was 7.5% each. Compared to the other two specimens, the presence of both $Al_2O_3$ and $B_4C$ was high in SP3. Additionally, the level of packing of the ceramic particles with the matrix material helped add more strength to SP3. The tensile strength of SP3 was 154 N/mm$^2$, which was more than that of the remaining two specimens. The increase in the ceramic content in the composite material increased the strength and decreased the elongation percentage. Similar results have been observed in the literature, where a decrease in the elongation percentage with an increase in the composition of the ceramic particles has been observed. There was an increase in the strength of the composite specimen with an increase in the wt.% of the ceramic particles. The most likely cause of this increase in tensile strength is hard particles, which can withstand a greater load when uniformly distributed in the matrix phase. The applied load is passed to the implanted hard particle, increasing the loading capacity of the produced composite. An increase of 12% and 23.2% of tensile strength has been observed with the increase in ceramic particle content of 10 wt.% and 15 wt.%, respectively, whereas a decrease of 25% and 37.5% of elongation in the composite sample with the increase in ceramic content to 10 wt.% and 15 wt.%, respectively, was observed in the present study.

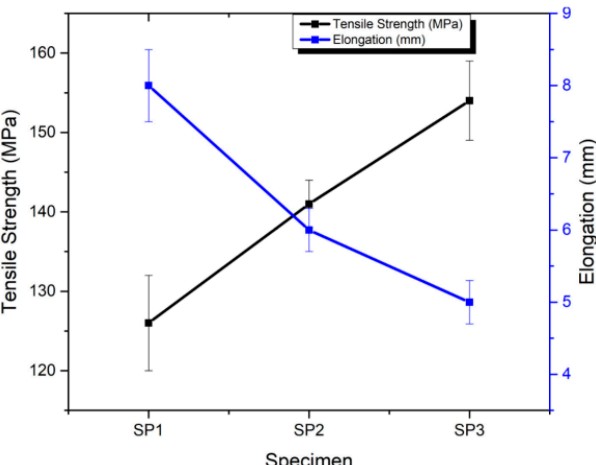

**Figure 4.** Tensile strength of the tested specimen.

## 4. Discussion

### 4.1. SEM of Fractured Specimens

SEM was used to identify the interfacial bonding and fracture behaviour of the composite specimens. In the present work, the three fractured specimens were investigated using SEM. The images obtained using SEM are shown in Figure 5. The graph of SP1, Figure 5a, which has lesser reinforcement content than the other composites, exhibited voids and ridges with closely spaced ceramic particles. The insufficient distribution of the

particles in a particular region resulted in the formation of voids. Figure 5b of specimen SP2 shows the formation of good bonding. This indicates that an increased wt% of ceramic particles to the metal matrix restricted the deformation of the matrix. The improvement in the bonding between the metal matrix materials with the reinforced ceramic particles improved the strength of the composites. From these images, it is also observed that the composite specimen, SP3, exhibits more bonding of ceramic particles. Thus, Figure 5a indicates the poor bonding nature of the composite samples which have a smaller percentage of ceramic particles. By contrast, in Figure 5b,c there is an increased wt% of ceramic particles, as evidenced by the good bonding between the ceramic particles and the matrix material. The formation of a particle cluster can be observed in Figure 5a, which is quite common in the MMCs due to improper bonding with the reinforcements, which will lead to the failure of the samples.

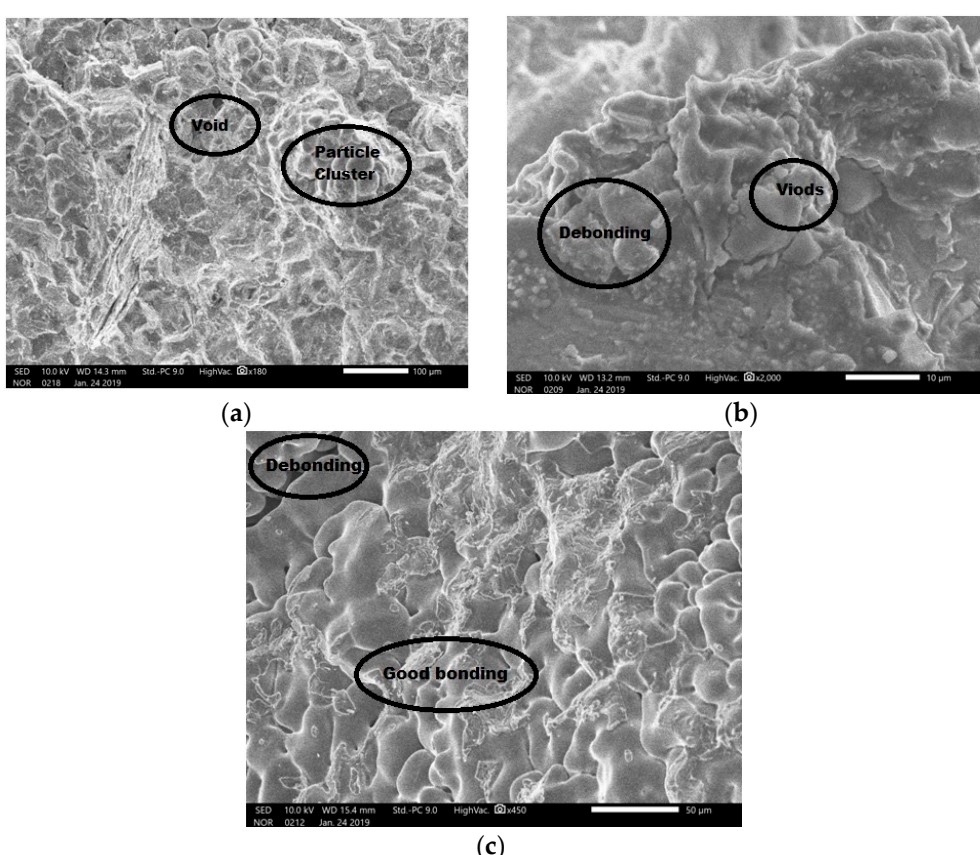

**Figure 5.** SEM images of (**a**) SP1, (**b**) SP2 and (**c**) SP3.

## 4.2. Effect of Load on Specific Wear Rate (SWR)

From the literature, it was evident that the fabrication of the composites affects the tribological properties [24]. The factors that affect the wear of the fabricated composite specimens were mainly the load imparted on the specimen and the size and the volume of the reinforcement particles [14]. The specific wear rate with respect to the load imparted on the specimen and the corresponding graphs are shown in Figure 6. It is evident from the graphs that there was an increase in the SWR with the increase in the proportions of the ceramic particles. A reduction in the SWR corresponding to the increase in the load was also observed. SP3, which had a higher wt% of both $B_4C$ and $Al_2O_3$, had a high SWR. Similar results have been experienced in the literature as well [13]. The ceramic material, $B_4C$, being the hardest material, increased the wear rate of the composites. Thus, the increase in $B_4C$ in the composition reduces the destruction of the fabricated composite and improves wear resistance, thereby decreasing the specific wear resistance. The reduction of wear loss of the composites can be associated with an improvement in hardness of manufactured

composites by means of the introduction of a harder phase. The wear resistance was less at maximum load condition for SP1, which had a smaller percentage of reinforcement. This shows that when the sliding speed is changed at a specific load, the wear rate increases. When compared to hardness, it was shown that wear loss follows a similar pattern. Another conclusion is that when a larger sliding speed is enforced, the wear loss becomes more pronounced. According to the findings, the quantity of wear loss is determined by the applied speed. Abrasive wear was evident from the SEM images (Section 4.5). The SWR is increased as the percentage of the reinforcement particles increased in the composite samples. The increased wear loss at 30 N normal load could be due to a physical process in which shear forces increase significantly, causing more interface material to be lost. Elkady et al. [34] tested the wear activity of Cu-based boron nitride composite materials that used a sliding speed of 0.2 m/s and varied typical loading. The findings revealed a progressive rise in wear loss, which was consistent with typical load fluctuation.

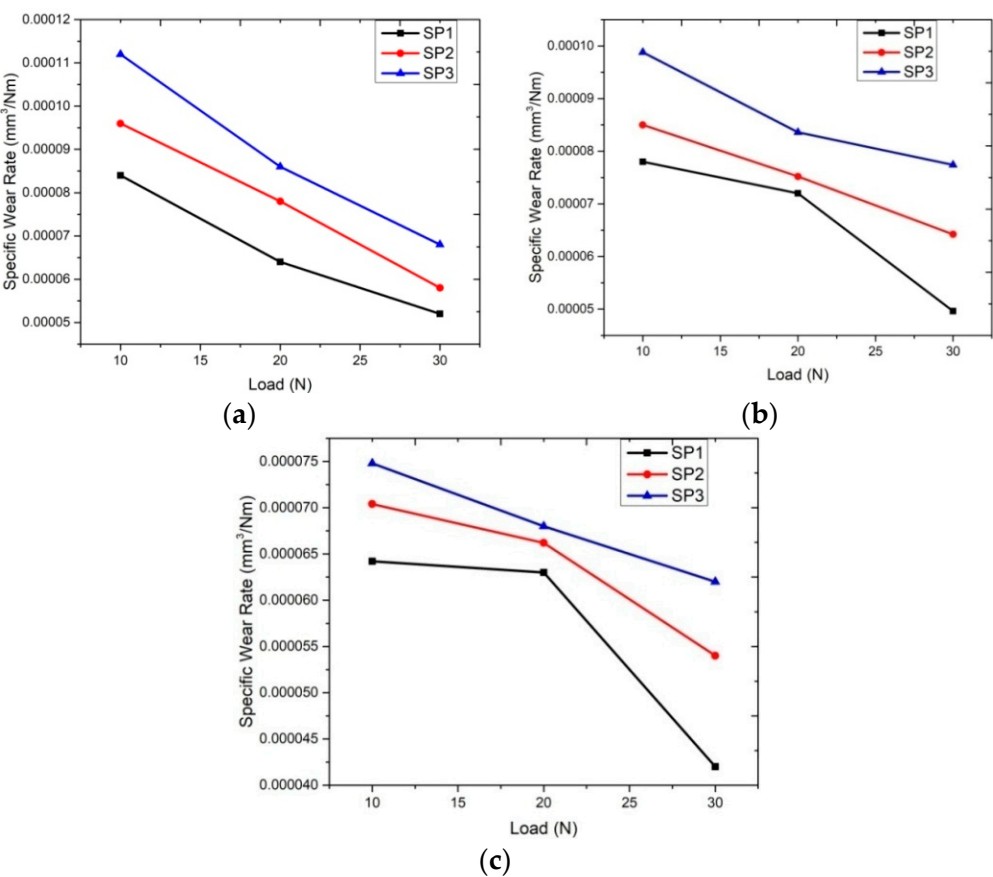

**Figure 6.** Influence of load on SWR at a speed of (**a**) 500 rpm, (**b**) 600 rpm and (**c**) 700 rpm.

### 4.3. Impact of Load on the Coefficient of Friction (CoF)

A pin on disk machine attached with the computer was used to display the data gathered from the tribometer equipped with frictional sensor. Figure 7 exhibits the dissimilitude of the coefficient of friction (CoF) with the load. With the increase in the load, the CoF decreases for all the three composite samples. Additionally, with the increase in the reinforcement content there was decrease in the CoF. This was evident from the previous studies also. The integration of hard particles and the good bonding ability at the surface of the matrix element with the tough particles could involve greater hardness characteristics and good fracture toughness of the composites, which might also promote a decrease in the friction coefficient. The increment in the wt% of this ceramic particle increases the hardness of the composite. The increase in the surface hardness of the composite decreases the CoF. Besides the enhancement of the reinforcement, the particle content decreases the

contact area of the disk as the pin slides against it. This results in the destruction of the disc material as it is exposed to the harder material with higher loads. Hence the CoF reduces. CaO-filled SnBr alloy composites were developed by Gangwar et al. [35]. The authors also simulated the wear performance in a dry environment and discovered the very same phenomenon as in the current investigation. The thin coating can be removed with just a little more exertion in a dry sliding state [36]. As a result, when the counter surface makes contact with the pin, this becomes smooth, resulting in greater friction coefficients.

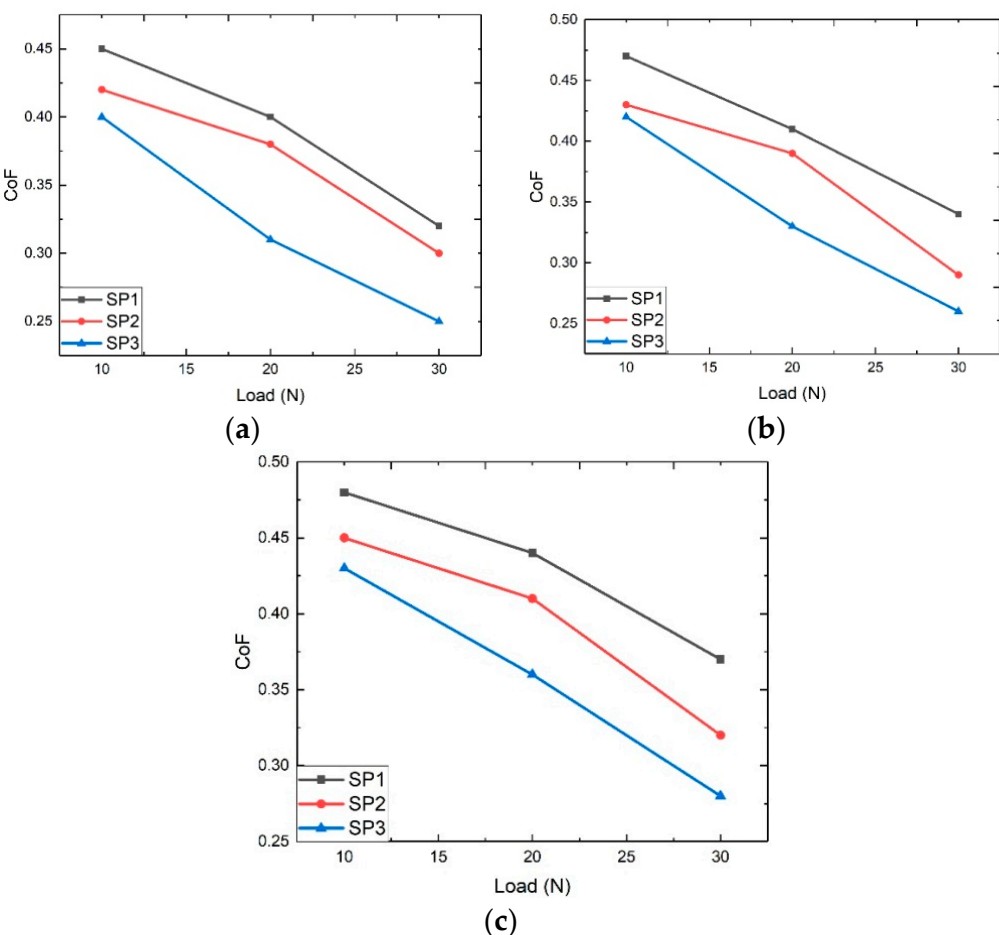

**Figure 7.** Influence of load on CoF at a speed of (**a**) 500 rpm, (**b**) 600 rpm and (**c**) 700 rpm.

### 4.4. Optical Microscope

Figure 8 shows the optical microscope image of the three fabricated composite samples. The white regions which are visible in the images are aluminium matrix, whereas the black regions are the micro-pores present in the composite samples. The porosity is very low in the SP3 sample as it can be observed in Figure 8c. Many pores are observed in the SP1 sample when compared to the SP3 and SP2 samples. Proposer stirring is one of the major causes for the porosity reduction in the composite samples. During the solidification of the composite samples, porosities are removed much as the molten metal falls from top to the bottom in the die, which removes all the gases so that the pores are eliminated [37]. The porosity obtained in the optical microscopy images correlates well with the microstructure.

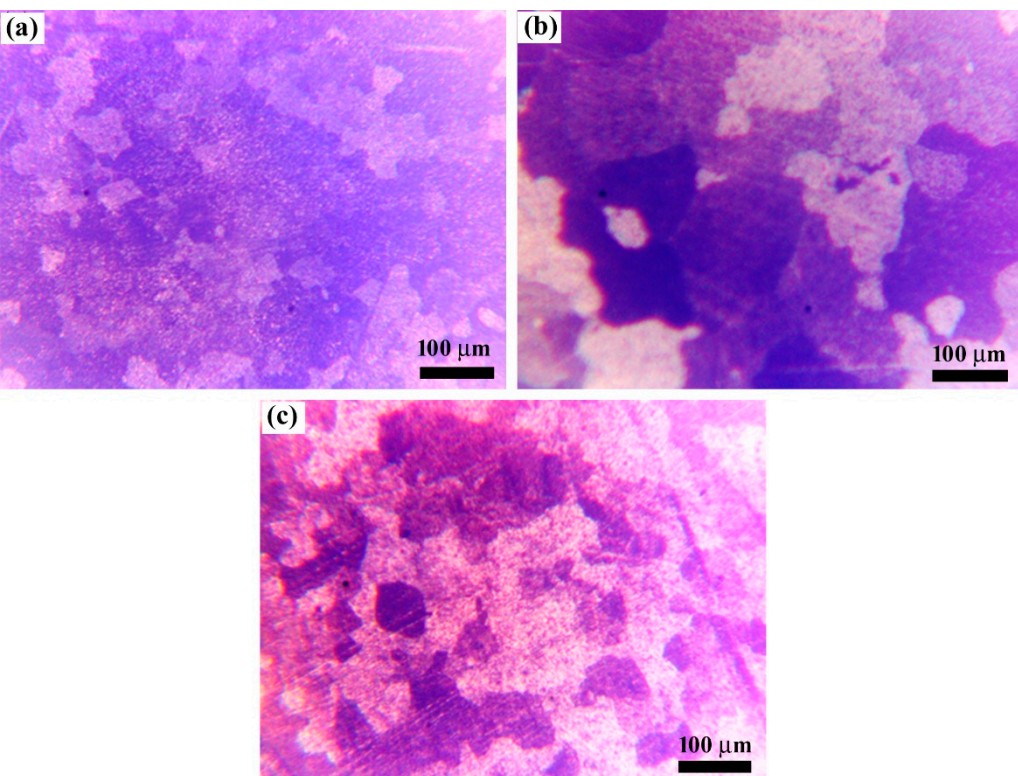

**Figure 8.** Optical microstructures of (**a**) SP1, (**b**) SP2 and (**c**) SP3.

*4.5. SEM of the Wear-Tested Specimens*

The characterization of the samples was investigated using SEM. The basic aim of the microstructure study on the wear-tested samples is to know the nature of the wear tracks. The images from SEM are shown in Figure 9. From the figures, the wear type observed was abrasive wear. The increase in the weight percentage of the ceramic particles resulted in an increase in the wear resistance of the specimen. This is depicted in Figure 8a–c. The SEM micrographs of all the three fabricated composite samples exhibit longitudinal grooves with some minute partial irregular pits; this indicates an adhesive type of wear in the composite samples. The presence of grooves in the wear-tested composite samples indicates the effect of micro cutting and micro ploughing on the counter-face, whereas the depths or prows are symbolic of adhesive failure of the Al 5059/$B_4C$/$Al_2O_3$ hybrid composite [38]. Figure 9c of SP3, which had a high content of $B_4C$ and $Al_2O_3$, shows a reduction in wedges and grooves as compared to Figure 9a, which had a high number of wedges and grooves. The surface microscopy following the loading condition displayed particular cracks coupled with continuous grooves when the quantity of reinforcements was increased. The continuous ridges are most likely the source of the asperities' strong absorption. The abrasion activity in connection to fractured hard reinforcement particles which serve as third layer abrasions could promote action. The surface of a unique composite containing higher reinforcement content shows the production of a complicated wear process that involves macro fractures and debonding wear patterns, and even some segregation particles. A specimen with a lesser amount of reinforcement particles exhibited less wear resistance. Figure 9b exhibits a wedge line along with clusters and voids. This shows a lesser wear resistance compared to SP3. More wear resistance was observed in SP3 when compared to SP2. Ceramic particles restricted the wear behavior of the composite samples; the higher the number of ceramic particles, the higher the wear resistance, since the ceramic particles restrict the composite sample from deformation, which increase the sample's hardness. Hard material is prone to wear resistance. The lower wear loss is due to a higher reinforcement wt% content, which provides a stronger adhering mechanism between the base matrix and the hard reinforcement particles. As a result, this composition has better hardness

characteristics. The utilization of composite materials has increased recently due to the higher wear resistance especially in the tribological applications.

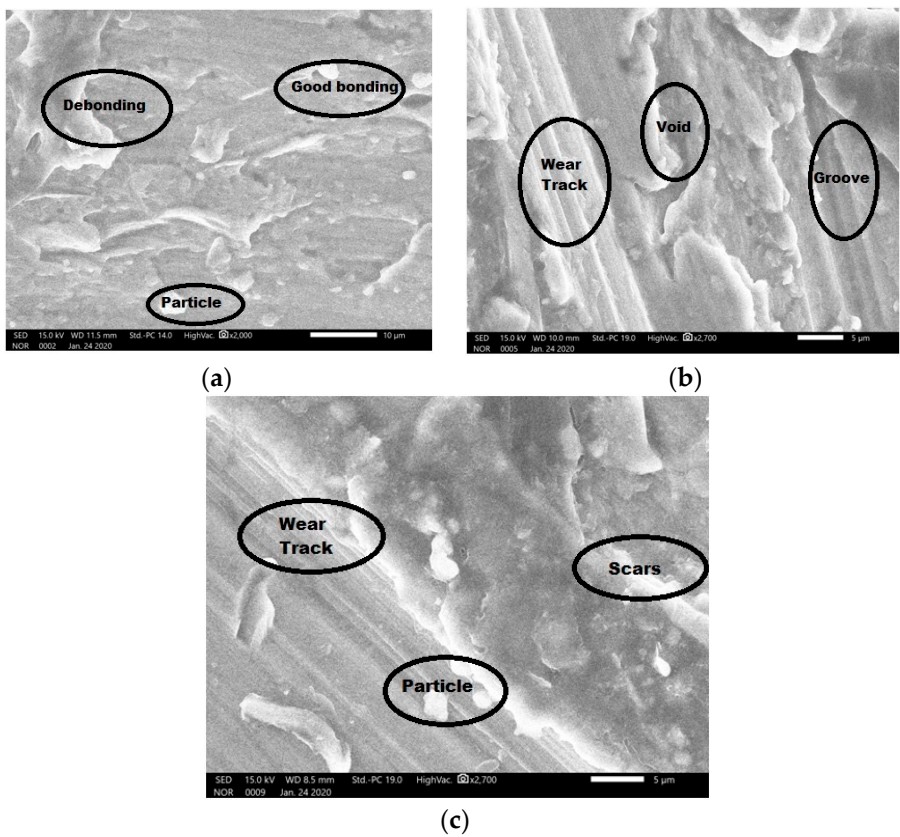

**Figure 9.** SEM images of wear samples (**a**) SP1, (**b**) SP2 and (**c**) SP3.

## 5. Conclusions

In the present study, a matrix material, Al 5059, was mixed with equal proportions of ceramic materials, $B_4C$ and $Al_2O_3$. The wt% of the combined ceramic materials considered was 5%, 10% and 15%. Using the stir casting method, three specimens were fabricated. The tested specimens of HMMC exhibited improvement in the mechanical properties of the metal matrix with an increase in the reinforcement content. The following observations were drawn from the wear test and SEM analysis.

1.    There is an increase in the wear resistance of the HMMC specimens with an increase in the wt% of ceramic particles.
2.    The SWR of the fabricated specimens decreased with an increase in the load on the specimens. The SWR of SP3 at 30 N load was observed to be $65 \times 10^6$ mm$^3$/Nm.
3.    The CoF decreases with an increase in the wt% percentage of the ceramic particles in the fabricated specimens. The increase in the load also leads to a decrease in the CoF of the HMMC specimens. The CoF of SP3 was observed to be 0.27 at 30 N load and a speed of 700 rpm.
4.    A consistency in the distribution of the ceramic particles has been identified from the SEM images of the fractured specimens under tensile testing.
5.    A virtual decrease in the existence of voids and grooves has been determined from the SEM images.

**Author Contributions:** Conceptualization, P.V.R. and S.V.; methodology, U.P.; formal analysis, P.V.R. and U.P.; investigation, P.V.R.; resources, U.P., S.V. and M.C.; data curation, U.P. and M.C.; writing—original draft preparation, U.P.; writing—review and editing, P.V.R. and S.V.; visualization, M.C.; supervision, P.V.R. All authors have read and agreed to the published version of the manuscript.

**Funding:** This research received no external funding.

**Institutional Review Board Statement:** Not applicable.

**Informed Consent Statement:** Not applicable.

**Data Availability Statement:** The related data as discussed in this article can be requested from the corresponding author P.V.R. or M.C.

**Conflicts of Interest:** The authors declare no conflict of interest.

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
