# Peer review of "Evaluation of Mechanical and Wear Properties of Al 5059/B4C/Al2O3 Hybrid Metal Matrix Composites"

_jcs, doi:10.3390/jcs6030086_

Round 1

Reviewer 1 Report

The manuscript is devoted to the study of mechanical properties and wear resistance of hybrid materials based on aluminum alloy, simultaneously reinforced with Al2O3 and B4C ceramic particles. The paper can be published in the journal but needs to be improved.

1) It is necessary to carefully proofread the manuscript, removing inaccuracies and errors. For example, the presence of line 21 raises questions.

2) The Introduction section is written very vaguely. In the Introduction, authors should either concentrate only on the consideration of works that are directly related to the system under study or, conversely, expand the circle of systems under consideration. Then, for example, why are not considered MWCNT-hybrid reinforcing particles used to improve the mechanical and tribological properties of aluminum matrix composites:

https://doi.org/10.1016/j.powtec.2018.03.023

https://doi.org/10.1016/j.msea.2020.139783

https://doi.org/10.1016/j.ceramint.2020.04.264 etc…

3) It is not clear from the text of the paper what size of ceramic particles was used to obtain hybrid materials. It is necessary to provide SEM images of ceramic particles, characterizing their average size and morphology.

4) Tribological properties were measured at different test speeds. In this case, it is more appropriate to use a linear velocity that does not depend on the radius of rotation, which is not given in the paper.

5) Data on the characterization of the structure of hybrid materials should be added to the paper. To do this, authors can use at least the method of optical microscopy.

6) It is difficult to precisely identify ceramic particles on SEM images. Point EDS data or EDS mapping images should be provided.

7) To understand the processes in the friction zone, data on the change in the chemical composition of the contact surfaces should be given, as, for example, it was done in the works.

https://doi.org/10.3103/S1068366620030022

https://doi.org/10.1134/S1027451021060410

etc...

8) Quantitative data characterizing the observed regularities should be added to the conclusions.

Author Response

Reviewer 1

The manuscript is devoted to the study of mechanical properties and wear resistance of hybrid materials based on aluminum alloy, simultaneously reinforced with Al2O3 and B4C ceramic particles. The paper can be published in the journal but needs to be improved.

  • It is necessary to carefully proofread the manuscript, removing inaccuracies and errors. For example, the presence of line 21 raises questions.

Author’s Response: Modified the mistake as per the suggestion.

2) The Introduction section is written very vaguely. In the Introduction, authors should either concentrate only on the consideration of works that are directly related to the system under study or, conversely, expand the circle of systems under consideration. Then, for example, why are not considered MWCNT-hybrid reinforcing particles used to improve the mechanical and tribological properties of aluminum matrix composites:

https://doi.org/10.1016/j.powtec.2018.03.023

https://doi.org/10.1016/j.msea.2020.139783

https://doi.org/10.1016/j.ceramint.2020.04.264 etc…

Author’s Response: All the above-suggested works are cited in the introduction as per the suggestion.

3) It is not clear from the text of the paper what size of ceramic particles was used to obtain hybrid materials. It is necessary to provide SEM images of ceramic particles, characterizing their average size and morphology.

Author’s Response: The facility which we used for SEM doesn’t contain the one which you are asking for. As per your suggestion optical microscope images are added in the revised article.

4) Tribological properties were measured at different test speeds. In this case, it is more appropriate to use a linear velocity that does not depend on the radius of rotation, which is not given in the paper.

Author’s Response: The authors worked only on the effect of load and speed on the weight loss of the article other than these, no other parameter is considered. Your suggestion may be taken up for our future works and we thank very much for your suggestion.

5) Data on the characterization of the structure of hybrid materials should be added to the paper. To do this, authors can use at least the method of optical microscopy.

Author’s Response: Added as per your suggestion.

6) It is difficult to precisely identify ceramic particles on SEM images. Point EDS data or EDS mapping images should be provided.

Author’s Response: Optical microscopy images are added in the revised version of the article, as of now, its not possible for including SEM images with EDS mapping. Your suggestion may be taken in our future works.

7) To understand the processes in the friction zone, data on the change in the chemical composition of the contact surfaces should be given, as, for example, it was done in the works.

https://doi.org/10.3103/S1068366620030022

https://doi.org/10.1134/S1027451021060410

Author’s Response: Added as per the suggestion

8) Quantitative data characterizing the observed regularities should be added to the conclusions.

Author’s Response: Added as per the suggestion.

Reviewer 2 Report

1. This is an interesting article describing current research problems.
2. "Introduction" section quite long, but well done in my opinion.
3. The reference to the manufacturing procedure is good practice, however I would insist on introducing a few sentences about manufacturing or add the manufacturing chart.
4. The research on mechanical properties and breakthroughs is very interesting, but I would like to see the microstructure of these castings. Are the reinforcement particles evenly distributed? It's quite hard to obtain. How the researchers dealt with it.
5. Please mention a few words about the recycling of potential aplication of this material.
6. Please expand your conclusions. Please find the causes and dependencies on the properties obtained.
7. In my opinion, microscopic examination of sections is required. 
8. Please, make the work checked in terms of English

Author Response

  1. This is an interesting article describing current research problems.

Authors response: Thank you for your positive feedback on our submission.

  1. "Introduction" section is quite long but well done in my opinion.

Authors response: Thank you.

  1. The reference to the manufacturing procedure is good practice, however I would insist on introducing a few sentences about manufacturing or add the manufacturing chart.

Authors response: Thank you for your valuable suggestion, it has been added to the revised manuscript.

  1. The research on mechanical properties and breakthroughs is very interesting, but I would like to see the microstructure of these castings. Are the reinforcement particles evenly distributed? It's quite hard to obtain. How the researchers dealt with it.

Authors response: Authors have used stir casting method to distribute the particles evenly into the matrix materials along with ultrasonication.

  1. Please mention a few words about the recycling of potential application of this material.

Authors response: Thank you. It was added to the revised manuscript.

  1. Please expand your conclusions. Please find the causes and dependencies on the properties obtained.

Authors response: Conclusions are updated as per the suggestion.

  1. In my opinion, a microscopic examination of sections is required. 

Authors response: Thank you.  Microstructures of the three types of studies were added to the revised manuscript.

  1. Please, make the work checked in terms of English

Authors response: English and Grammar corrections were done thoroughly for the entire manuscript.

Round 2

Reviewer 1 Report

.

Reviewer 2 Report

The authors responded to the comments on a minimalistic level. However, they answered every question. 

This manuscript is a resubmission of an earlier submission. The following is a list of the peer review reports and author responses from that submission.

Round 1

Reviewer 1 Report

Good afternoon, respected editor of the journal and colleagues!

I bring to your attention my opinion about the article: Evaluation of Mechanical and Wear Properties Of Al 5059/B4C/Al2O3 Hybrid Metal Matrix Composites.

- This article discusses in evaluation of Mechanical and Wear Properties Of Al 5059/B4C/Al2O3 Hybrid Metal Matrix Composites. In the review of the literature, various types of composites with an aluminum matrix are considered in sufficient detail, however, the very rapidly developing composites with a titanium matrix are not considered at all as a comparison. Therefore, I think it would not be superfluous to consider the following works in the literature review:

Morsi, K. Review: Titanium–titanium boride composites.  J. Mater. Sci.

Brittle-to-ductile transition in a Ti–TiB metal-matrix composite.

M.Y. Koo, J.S. Park, M.K. Park, K.T. Kim, S.H. Hong, Effect of aspect ratios of in situ formed TiB whiskers on the mechanical properties of TiBw/Ti-6Al-4V composites, Scr.Mater. 66 (2012) 487–490.

«Effect of Hot Rolling on the Microstructure and Mechanical Properties of a Ti-15Mo/TiB Metal-Matrix Composite»

Leyens, C.; Peters, M. Titanium and Titanium Alloys. Fundamentals and Applications; Wiley-VCH: Weinheim, Germany, 2003; pp. 1–499. 

Chen, B.; Shen, J.; Ye, X.; Jia, L.; Li, S.; Umeda, J.; Takahashi,M.; Kondoh, K. «Length effect of carbon nanotubes on the strengthening mechanisms in metal matrix composites».

«Microstructure evolution of a Ti/TiB metal-matrix composite during high-temperature deformation»

Saito, T.; Furuta, T.; Yamaguchi, T. Development of low cost titanium matrix composite. In Advances in Titanium Metal Matrix Composites, the Minerals, Metals and Materials Society; Froes, F.H., Storer, J., Eds.; TMS: Warrendale, PA, USA, 1995; pp. 33–44.

Laser Beam Welding of a Ti-15Mo/TiB Metal–Matrix Composite

In the attached version of the article, only the first and second figures are present, the rest are not! Please attach a revised version of the article with an extended overview part with added analyzes of the above literature sources and all the figures.

I believe that with the addition of the above analysis of the literature, this manuscript will become more attractive.

After correcting above comments, the article will be revised again.

Reviewer 2 Report

The article has the following comments.

  1. There are many grammatical typos and errors. For example, no spacing between words in lines 83, 100, 108, etc. For some unknown reason, in the title, some of the words are written with a capital letter, and some with a small one. In line 8, instead of "interest of efficient materials", it should be "interest in efficient materials". There are also other grammatical errors. Sometimes phrases are difficult to understand. For example, in lines 40, 42 we read "The composites made from particulate metal matrix comprise of isotropic properties, then composites reinforced of long fiber". So, the level of English is too low.
  2. One sentence often contradicts another. For example, in the abstract, in the penultimate sentence we read "... there is decrease in the percentage of elongation of the specimen", while the last one says " An increase in the ceramic content in the composite samples made the composite sample ductile from brittle". Moreover, there are contradictions between the authors' conclusions and the experimental data obtained. In lines 254, 255 we read "... the increase in the composition of B4C ... improves the wear resistance". However, in Fig. 6 we see the opposite situation - with the increase in content of ceramics, the wear rate increases. 
  3. In lines 125, 126, instead of describing the method of obtaining samples, reference is made to the works [24-26]. This is unacceptable, it is necessary to at least briefly describe the method of sample manufacturing and the equipment that was used. 
  4.   Fig. 1 shows only two samples, although three were made. Sample sizes are not specified. When looking at the photos in Fig. 1, you see that the first sample is flat, and the second one looks like cylindrical. However, when we look at Fig. 2, all three samples are cylindrical. The question arises, from where and what samples were taken, and who made them and how.
  5. In line 144 it is written that the hardness was measured according to the ASTM E384 standard and the reference [27] is indicated. But, reference [27] says that this is a standard for measuring microhardness, not hardness. The question arises on which device was the Brinell hardness measured. 
  6. The name of the tribometer on which the wear resistance test was made is not specified. Table 3 is constructed incorrectly, as it can be understood that all three samples were tested under different conditions. 
  7. A serious disadvantage of the work is that additional sample was not made from pure Al5059 without the addition of boron carbide and alumina. Comparison with the sample of pure Al5059 is very important to understand whether there is any gain in properties from ceramic additives at all.  The fact is that the commercial unhardened AL5059 alloy has a tensile strength of 330 MPa and a relative elongation of 24%. These parameters are much higher then those of the samples obtained by authors and shown in Fig. 4.
  8. The conclusions in lines 338, 339 and 342, 343 are completely unreasonable. How can it be stated the exhibited improvement in properties if samples from a pure Al5059 alloy have not been tested. And how can it be stated that with an increase in the content of ceramics wear resistance increases, when the diagrams in Fig. 6 show the opposite.             

Reviewer 3 Report

In the manuscript entitled "Evaluation Of Mechanical and Wear Properties Of Al 5059/B4C/Al2O3 hybrid metal matrix composites", the authors studied the mechanical properties of hybrid aluminum matrix composites. This manuscript lacks originality as the findings are known to the scientific society. The authors fail to explain why hybrid particles are needed. In line 102, it mentions boron carbide is for erosion resistance, but there is no data support. The experimental methods lack details, such as hardness data points and tensile test sample geometry. There is no description of the raw particle size and morphology, nor are there SEM pictures of the particle dispersion in the composite. Therefore, I suggest rejection.